Enhancing ransomware defense: deep learning-based detection and family-wise classification of evolving threats

Hussain Amjad 1
Saadia Ayesha ayesha.saadia@mail.au.edu.pk ayesha.saadia@students.au.edu.pk 2
Alhussein Musaed 3
Gul Ammara 4
Aurangzeb Khursheed kaurangzeb@ksu.edu.sa 3
1 Department of Cyber Security, Air University , Islamabad , Pakistan
2 Department of Computer Science, Air University , Islamabad , Pakistan
3 Department of Computer Engineering, College of Computer and Information Sciences, King Saud University , Riyadh , Saudi Arabia
4 Faculty of Computing, Engineering and the Built Environment, Birmingham City University , Birmingham , United Kingdom
Alatas Bilal
Electronic publication date: 2024 Nov 29
Publication date: 2024
Volume: 10
Electronic Location ID: e2546
Received 2024 May 30; Accepted 2024 Nov 5
Copyright: ©2024 Hussain et al.
Copyright year: 2024
Copyright holder: Hussain et al.
License: This is an open access article distributed under the terms of the Creative Commons Attribution License, which permits unrestricted use, distribution, reproduction and adaptation in any medium and for any purpose provided that it is properly attributed. For attribution, the original author(s), title, publication source (PeerJ Computer Science) and either DOI or URL of the article must be cited.
License URL: https://creativecommons.org/licenses/by/4.0/

Keywords: Ransomware detection, Ransomware classification, Ransomware family attribution, Artificial Intelligence, Deep learning, Machine learning

Funding: Research Supporting Project RSPD2024R553 King Saud University, Riyadh, Saudia Arabia This research is funded by Research Supporting Project Number (RSPD2024R553), King Saud University, Riyadh, Saudia Arabia. The funders did not have a role in study design, data collection and analysis or the decision to publish.

==============================
Ransomware is a type of malware that locks access to or encrypts its victim’s files for a ransom to be paid to get back locked or encrypted data. With the invention of obfuscation techniques, it became difficult to detect its new variants. Identifying the exact malware category and family can help to prepare for possible attacks. Traditional machine learning-based approaches failed to detect and classify advanced obfuscated ransomware variants using existing pattern-matching and signature-based detection techniques. Deep learning-based approaches have proven helpful in both detection and classification by analyzing obfuscated ransomware deeply. Researchers have contributed mainly to detection and minimaly to family attribution. This research aims to address all these multi-class classification problems by leveraging the power of deep learning. We have proposed a novel group normalization-based bidirectional long short-term memory (GN-BiLSTM) method to detect and classify ransomware variants with high accuracy. To validate the technique, five other deep learning models are also trained on the CIC-MalMem-2022, an obfuscated malware dataset. The proposed approach outperformed with an accuracy of 99.99% in detection, 85.48% in category-wise classification, and 74.65% in the identification of ransomware families. To verify its effectiveness, models are also trained on 10,876 self-collected latest samples of 26 malware families and the proposed model has achieved 99.20% accuracy in detecting malware, 97.44% in classifying its category, and 96.23% in identifying its family. Our proposed approach has proven the best for detecting new variants of ransomware with high accuracy and can be implemented in real-world applications of ransomware detection.

Introduction

The evolution of the internet has revolutionized our way of connecting, communicating, and accessing information but along with its amazing advancements come unexpected difficulties, especially in cybersecurity. With the internet growing in popularity and expanding its virtual network around the world, cybercriminals took advantage of the opportunity to profit from its accessibility and anonymity (Greubel, Andres & Hennecke, 2023). The inception of the World Wide Web (WWW) provided hackers with new avenues to infiltrate systems, and with the rise of interconnected devices, the potential for disruption grew exponentially (Garetto, Gong & Towsley, 2003). With the maturation of the digital landscape, cybercriminals employed increasingly sophisticated tactics, including deploying malware (a class of software designed to infiltrate, damage, or gain unauthorized access to systems and data) (Aslan & Samet, 2020). However, malware was just the beginning of the cybercriminals’ wicked inventiveness and it reached its zenith with the advent of ransomware (a malicious software that encapsulates the darkest aspects of cybercrime) (Richardson & North, 2017). In this research, we aim to develop and evaluate effective techniques for detecting and classifying ransomware, focusing on overcoming the limitations of traditional detection methods. We propose an advanced framework that enhances ransomware detection by leveraging deep learning-based approaches, particularly for newly discovered and obfuscated threats. This work contributes to improving the accuracy and robustness of modern antivirus systems.

Ransomware utilizes encryption techniques to lock victims’ files and data, holding them hostage until a ransom is paid (Aurangzeb & Islam, 2017; Kok, Abdullah & Jhanjhi, 2022). This evolution represented a seismic shift in cyber-attacks, combining extortion and encryption with potentially devastating consequences (Sharmeen et al., 2020). The progression from conventional malware to formidable ransomware was symbolic of the ongoing struggle between security experts and those who sought to exploit the vulnerabilities of an interconnected world (Beaman et al., 2021). The history of the internet is thus intertwined with both incredible innovation and the ever-present need to protect against the dark forces that seek to exploit its potential (Kargaard et al., 2018).

There are two types of ransomware; the first one is crypto ransomware, which works by encrypting the victim’s files and data and the second type is locker ransomware which locks or disables the victim’s computer access (Anghel & Racautanu, 2019). Locker ransomware is not dangerous if it is compared with crypto ransomware because it just disables the user’s accessibility to the device but the data on the device is usually unaffected (Gómez-Hernández, Álvarez-González & García-Teodoro, 2018). Removing infection results in access to the device and unchanged data. Relocating the storage device, generally, a hard drive, to another computer that is still in use will often recover data even if the virus is difficult to remove. In this case, the locker ransomware is less successful in demanding ransom payments (Subedi, Budhathoki & Dasgupta, 2018; Alhawi, Baldwin & Dehghantanha, 2018; Mohammad, 2020). Although ransomware has a long history in the late 1980s, cyber criminals used encryption techniques to exchange cash through postal services (Richardson & North, 2017). One of its examples was the AIDS ransomware, which was invented in 1989, and the victims had to send $180 to a P.O. box located in Panama to restore access to their systems (Paquet-Clouston, Haslhofer & Dupont, 2019). It was not well-known until 2009, but when Bitcoin was introduced in 2010, the rise of cryptocurrency encouraged cybercrimes since it gave offenders a safe, untraceable way to make payments without giving away who they were (Akbanov, Vassilakis & Logothetis, 2019). In 2013, the most notorious piece of ransomware, CryptoLocker, appeared, targeting Windows operating systems. It used cryptographic keys for encryption, employing a pair of public and private keys to encrypt and decrypt the victim’s files (Savenko et al., 2019). Over 300,000 systems were affected by a new WannaCry ransomware version in many countries in 2017 (Assegie, 2021). A total of 10,666 new ransomware variations were identified in the first half of 2022, according to research from FortiGuard in 2023. The likely cause of this shift is the mature operation of Ransomware-as-Service (RaaS), which assisted cybercriminals in introducing new and disguised ransomware versions (Report, 2023).

The researchers used different techniques to detect and classify malware while commercial antivirus companies widely depended on signature-based malware detection. Signatures of executable files are first extracted using static analysis and stored in a database. The signatures of a suspected sample file are then retrieved and compared to previously established signatures to classify malicious or benign. This comparison determines whether the sample file is malicious or not. While this approach is quick and effective for identifying existing malware, it is insufficient for detecting newly discovered malware. Moreover, malware from the same family can easily evade detection by using simple obfuscation techniques (Celdrán et al., 2023). With behavior-based malware detection, an application is identified as malicious or benign depending on how a program behaves. The behavior-based detection approach observes what the sample software does. Based on the behavior observed, the software sample is categorized into benign or malicious. The three steps of this technique are behavior extraction, property generation, and the implementation of machine learning models to determine the safety or benignity of the application under study. It uses system calls, application programmable interfaces (API) calls, modifications to files, registries, and computer networks to ascertain behavior. Using these sequences, behaviors are categorized, sequences are generated, and attributes are produced. Despite modifications to the software’s source code, the program’s functionality does not change. Consequently, various dangerous software variations are identified, and the majority of new malware is discovered using this approach (Muzaffar et al., 2022). Some malware programs, however, do not operate effectively in a protected environment and may be classified falsely (Palša et al., 2022). Another important factor is that classification received less attention compared to detection, and the majority of researchers have primarily focused on detection with limited work in classification. Accurate classification of ransomware families is crucial for effective mitigation and prevention (Razaulla et al., 2023).

With the advancement of machine learning (ML) techniques, many researchers have utilized them to identify ransomware. ML-based ransomware detection algorithms can model more complicated data patterns than traditional signature-based ones (Rawson & Brito, 2023). This allows them to successfully detect new strains of old malware, including previously undisclosed malware (Alarfaj et al., 2022). In general, there are three types of machine learning algorithms: unsupervised, semi-supervised, and supervised (Parlett-Pelleriti et al., 2023). In supervised learning, a prediction model is trained on preexisting labeled data to predict the labels of incoming data precisely (Yan & Wang, 2022). Conversely, unsupervised learning utilizes unlabelled data. In contrast, labeled and unlabeled data are combined with semi-supervised learning during the training phase (Kim & Lee, 2022). In supervised learning, it is challenging for the models to detect ransomware that has never been identified or is not from known families on which they have been trained. However, these types of models in general have fewer challenges and can detect ransomware more easily (Tayyab et al., 2022; Ali & Ullah, 2022). Traditional ML models normally need human interaction to work because they cannot analyze raw input data (Moshayedi et al., 2022).

Deep learning (DL), a new branch of Artificial Intelligence (AI) and a type of ML has gained much attraction and popularity and become a primary machine learning method in different fields. These models utilize artificial neural networks (ANNs) and learn from making use of different hidden layers, and from previous examples. Many layers of synthetic neurons are included whose weights are continuously adjusted to get the intended results. These adjustments and modifications are made to make sure that the optimizer may minimize the loss. The loss is the prediction error of the ANN and can be calculated by a loss function (Zhang, Liu & Jiang, 2020). Recurrent neural networks (RNNs), multilayer perception (MLP), and convolution neural networks (CNNs) are the most popular DL designs (Khan et al., 2020). DL models have complex architectures and many processing layers, they can learn deeper features automatically with varying degrees of abstraction and can handle high-dimensional data. More research is needed to evaluate deep learning properly, which has produced highly accurate results in many fields but has not yet been implemented widely in ransomware detection, category-wise classification, and family attribution (Ficco, 2021).

To that purpose, this research presents an RNN-based group normalization-based bidirectional long short-term memory (GN-BiLSTM) model to not only detect ransomware with high accuracy but also classify its category and family by integrating different behavioral variables, derived from dynamic analysis. To verify the applicability of the proposed model in a real-world scenario, 10,876 samples of 26 malware families including 11 ransomware families and 10,876 benign samples are collected, analyzed dynamically, and prepared for machine learning in comma-separated-values (CSV) format. API calls, logs, file auditing, registry access, and network traffic are among the behavioral characteristics collected through dynamic analysis. The findings reveal that the proposed model efficiently identifies, classifies, and assigns unknown ransomware variants to families. This study makes the following contributions:

• The first contribution is to propose a DL-based method that can detect the latest obfuscated ransomware variants with high accuracy.

• The second contribution is to perform multi-class classification to detect malware categories or their family.

• The final contribution is to collect the latest ransomware samples, create a dataset, train models on it, and validate performance.

This paper is further organized in such a way that the second section provides a summary of previous research. The experimental design methodology and implementation of the GN-BiLSTM model are provided in the third section. In the fourth section, practical implementation and results are analyzed, discussed, and compared with related work. The fifth section provides the conclusion and outlines potential future research directions based on this study.

Related Work

Researchers have used different techniques and approaches in the literature to detect ransomware. We conducted a literature review by identifying the motivations, problems addressed, proposed solutions, datasets used, features extracted or selected, trained models, reported accuracy, limitations, and especially multi-class classification performed or not for each paper.

Machine learning

Several researchers have used typical machine learning-based approaches to identify ransomware. Zhang, Liu & Jiang (2020) suggested a method for analyzing the relationship between characteristics and malware labels using soft relevance assessment. The authors used the Microsoft Malware Classification Challenge (BIG 2015) dataset, training models including naïve Bayes (NB), decision tree (DT), Random Forest (RF), and support vector machine (SVM), and achieved an overall accuracy of 98.8%. Khan et al. (2020) used a digital DNA sequencing-based approach called linear regression (LR), RF, NB, and sequential minimal optimization (SMO) to achieve 87.9% accuracy on a dataset of 582 ransomware and 942 benign samples. Another noteworthy study, Ficco (2021), presented a weighted average technique called Alpha-Count, which included antiviral, intrusion detection systems, file entropy, and network traffic analysis. The deep neural network (DNN) model, trained on 10,634 malicious and 2,000 benign samples, attained an accuracy of 93.28%.

Similarly, Poudyal & Dasgupta (2021) evaluated ransomware activity using system calls, network traffic, and file actions, training machine learning models (SVM, LR, and RF) with SVM attaining 99.72% accuracy. Mail, Ab Razak & Ab Rahman (2022) suggested a cloud-based sandboxing system that used machine learning algorithms (RF, J-48, and NB) to achieve 99.8% accuracy on a dataset of 9,600 malware samples. Ganfure et al. (2023) created RTrap, a dynamic analytic tool that detects ransomware in a controlled setting and achieves 99.8% accuracy using models (DT, RF, and SVM). Molina et al. (2021) used a technique for ransomware family attribution by analyzing the pre-attack behavior of ransomware. A total of 129,500 malware samples were collected from VirusTotal and VirusShare from 2010 to 2019, of which 19,499 samples belonged to 21 ransomware families. Bernoulli BN, K-nearest neighbor (KNN), ANN, long short-term memory (LSTM), and RF models were implemented. The RF model achieved a maximum of 94.92% accuracy.

Deep learning

Deep learning algorithms have also shown substantial effectiveness in malware detection. Zhang, Wang & Zhu (2021) suggested a dual GAN-based technique for distinguishing encrypted and unencrypted files, obtaining 98.1% accuracy on the KDD99, SWaT, and WADI datasets. Aslan & Yilmaz (2021) used AlexNet and ResNet152 on the Microsoft BIG 2015, Malimg, and Malevis datasets, obtaining 97.78%, 94.88%, and 96.5% accuracy, respectively. Li, Rios & Trajković (2022) adopted a different strategy, evaluating border gateway protocol (BGP) routing data using models including LightGBM, CNN, and RNN, and achieved an accuracy of 64.74%. Darem et al. (2021) introduced an adaptive malware detection algorithm based on the Drebin Android malware dataset that achieved 99.41% accuracy. Yazdinejad et al. (2020) used the LSTM-based approach for malware detection with 10-fold cross-validation. The model achieved 98% accuracy in detecting malicious samples from benign. Hwang et al. (2020) proposed a DNN-based malware detection technique and trained the model on a dataset of 10,000 malicious and 10,000 benign samples. Their proposed approach achieved 94% accuracy.

Recently, researchers have looked at multi-class classification for malware family detection. Roy & Chen (2021) employed a BiLSTM-CRF-based approach to identify ransomware and categorize anomalous events with 99.87% and 96.5% accuracy respectively. Keyes et al. (2021) introduced EntropyLyzer, an entropy-based behavioral analysis tool that classified 147 families and 12 categories of Android malware with 98.4% accuracy. Lashkari et al. (2018) created a new Android malware dataset (CIC-AndMal2017) and used KNN to identify malware, obtaining 85.4% accuracy; however, the model struggled with family attribution (27.24%). Rahali et al. (2020) used deep learning to classify malware into 12 categories and 191 families, with 93.36% accuracy.

The CIC-MalMem-2022 dataset

Obfuscated malware detection has been a critical challenge addressed by various researchers. Carrier (2021) used the VolMemLyzer framework with 26 memory features, achieving 99% accuracy across several classifiers like SVM, DT, RF, and KNN. Smith, Khorsandroo & Roy (2023b) employed seven classification algorithms and achieved 99% accuracy using Pearson correlation. Naeem et al. (2023) used the Malware-Exploratory dataset to apply clustering methods (K-Means, DBSCAN, and GMM) and seven more classifiers, reaching an average of 99% accuracy. Mezina & Burget (2022) used dilated CNNs for multi-class classification and achieved 99% accuracy. Roy et al. (2023) proposed MalHyStack, a multi-class classification model based on stacked ensemble learning, which achieved 99.98%, 85.04%, 70.20% accuracy in detection, classification, and family attribution, respectively. Dang (2022) used CatBoost to classify obfuscated malware and performed binary and multi-class classification. The model achieved 99.9% accuracy in detection. Dener, Ok & Orman (2022) performed binary classification using RF, DT, GBT, LR, NB, linear vector support machine, MLP, deep feed forward neural network, and LSTM. The highest accuracy achieved by the LR algorithm with 99.97%. In Al-Qudah et al. (2023), one class SVM (OCSVM) classifier was proposed with Component Analysis (PCA), and 99.4% accuracy in one-class classification using the PCA (OCC-PCA) model was attained.

Abualhaj et al. (2024) proposed an improved distance metric parameter of the KNN with K-fold cross-validation for malware detection. They achieved 99.97% accuracy in detection, 82.21% in classification, and 66.93% in family attribution. Shafin, Karmakar & Mareels (2023) proposed a CNN-BiLSTM-based approach, namely CompactCBL (Compact CNN-BiLSTM) and RobustCBL (Robust CNN-BiLSTM) to detect the binary attack, its type, and family. Methods achieved 99.96% and 99.92% accuracy in detecting binary attacks, 84.56% and 84.22% in family-wise attacks, and 72.60% and 71.41% in types of attacks, respectively. Smith, Khorsandroo & Roy (2023a) used a CNN-based stacked ensemble method as a base learner and MLP for meta-learning. IoT-based malware was identified and categorized by identifying image characteristics and detecting suspicious activities. It helped to classify malware families and the models achieved 99.01% accuracy in malware detection.

Many existing ransomware detection approaches rely primarily on handmade characteristics or static signatures, making them less adaptable to new and emerging malware families. Polymorphic and metamorphic ransomware strains are particularly difficult to identify using signature-based methods. A large majority of research concentrates on static or short-term behavioral characteristics, file actions, or network traffic over a given period. However, ransomware frequently demonstrates dynamic tendencies that evolve. Approaches that ignore these temporal patterns fail to capture the entire range of ransomware operations, perhaps leading to more false negatives. Many machine learning algorithms employed in prior studies perform well on known datasets but suffer when exposed to novel malware. This problem results from overfitting to specific patterns in the training data, reducing the model’s capacity to generalize across different ransomware families and innovative attacks.

Our literature analysis process leads us to the conclusion that it’s critical to identify obfuscated ransomware by analyzing it deeply, classifying its correct category (ransomware, Trojan, spyware, etc.), and attributing its families using advanced DL-based techniques to make defense ready by providing correct solution and path. Therefore, in this research, we proposed the GN-BiLSTM, a DL-based method to detect, classify, and attribute ransomware families. Unlike previous techniques that rely on predetermined characteristics, GN offers a flexible way to represent the links between various system activities (e.g., file operations, network activity, and registry changes) using a graph structure. This enables the extraction of rich, interdependent data that can capture the complex, multi-step operations undertaken by ransomware, resulting in improved detection of sophisticated threats.

The Proposed Methodology

This section provides the methodology for our proposed GN-BiLSTM model for ransomware detection, category-wise, and family-wise classification framework. The methodology of the proposed framework is presented in Fig. 1, comprised of 6 steps; datasets, data pre-processing, data balancing, feature selection, normalization, and DL implementation. Algorithm 1 provides a hierarchical view of the proposed methodology.

Figure 1 The proposed methodology.

____________________________ Algorithm 1 The Proposed Methodology for Ransomware Detection and Multi-class Classification   1:  Input:        ⋅  Two datasets:               – CIC-MalMem-2022:             – Self-Created Dataset:                  *  Collected ransomware samples.                    *  Analyzed using Cuckoo Sandbox.                    *  Generated JSON reports.                    *  Extracted features and compiled into CSV.   2:  Step 1: Data Preparation        ⋅  Prepare datasets for multi-class classification.   3:  Step 2: Data Cleaning        ⋅  Handle missing values and remove duplicated records.          ⋅  Perform categorical encoding.   4:  Step 3: Outlier Handling        ⋅  Identify and address outliers.   5:  Step 4: Data Balancing        ⋅  Apply data balancing techniques to address the class imbalance.   6:  Step 5: Feature Selection        ⋅  Use feature selection techniques to choose relevant features.   7:  Step 6: Data Normalization        ⋅  Normalize data using Min-Max scaling or Z-score normalization.   8:  Step 7: DL Model Implementation        ⋅  Implement six DL models:               – CNN, MLP, LSTM, CNN-LSTM, CNN-BiLSTM               – GN-BiLSTM (proposed model)          ⋅  Train and test each model.          ⋅  Evaluate performance using accuracy, precision, recall, F1-score, and AUC.   9:  Step 8: GN-BiLSTM for Ransomware Detection and Family Attribution        ⋅  Construct graph-based feature representations.          ⋅  Apply GN-BiLSTM to capture sequential dependencies and feature relationships.          ⋅  Output classification and family attribution results. 10:  Output: Detection results, classification labels, and family attribution.______________________________

Datasets

Two benchmark datasets are used in this research. The first one is OMM-2022, an obfuscated malware dataset (Carrier, 2021), and the second one is a self-created ransomware dataset.

The CIC-MalMem-2022 (OMM-2022) dataset

This is the latest obfuscated malware dataset generated in 2022 by the Canadian Institute for Cybersecurity. It comprises 58,596 records, including 50% malicious and 50% benign with 56 features and one class. The dataset contains three malware categories (Trojan Horse, Spyware, Ransomware) and 15 malware families. Table 1 provides sample information on the malware categories and their families.

Table 1 Malware categories and family information.

Category	Family	Samples	Total samples	
Trojan Horse	Zeus	195	9,487	
Emotet	196	
Refrose	200	
Scar	200	
Reconyc	157	
Spyware	180Solutions	200	10,020	
Coolwebsearch	200	
Gator	200	
Transponder	241	
TIBS	141	
Ransomware	Conti	200	9,791	
Maze	195	
Pysa	171	
Ako	200	
Shade	220	
Benign			29,298	

Self-created dataset

Samples are collected from multiple online resources to obtain real instances of malware. A total of 21,752 samples (10,876 malicious and 10,876 benign), including 11 ransomware families (Cerber, DarkSide, Dharma, GandCrab, LockBit, Maze, Phobos, REvil, Ragnar, Ryuk, Shade, WannaCry) are collected from VirusTotal, VirusShare, MalwareBazar, and GitHub. To further check the robustness of models, samples of three more malware categories (Trojan Horse, Info Stealer, RAT) are included in the dataset. Cuckoo Sandbox 2.0.7 was installed and configured with a Windows 7 (32-bit) operating system to analyze dynamic behavior. The malware samples are run in a controlled virtual box environment to capture the activities of malware samples during their execution. The activities include processes it started, files and registry activities it created, files read, or deleted, dynamic link libraries (DLLs), and application programmable interfaces (APIs) calls it made. The main focus was to capture API calls by malware, files opened, created, or deleted, and registry entries made. These actions indicate the malware’s behavioral dynamics. Cuckoo Sandbox produced an extensive JavaScript Object Notation (JSON) report. JSON files are then parsed and analyzed family-wise, and 51 features are extracted from each sample report using Python language script, labeled, and converted into a CSV file format.

Preprocessing

ML and DL models require preprocessed data to perform efficiently. Pre-processing is accomplished in many ways, and different techniques are applied to make datasets work with ML and DL models. In this research, the datasets are pre-processed through nine steps. In the first step, the datasets are prepared for detection and multi-class classification. In the second step, missing values and duplicated records are handled. In the third step, categorical encoding is performed, and in the fourth step, similarity and outliers are addressed. In the fifth step, data balancing is applied and in the sixth step, feature selection is conducted. In the seventh step, normalization is performed, and finally, in the eighth and ninth steps, deep learning (DL) models are trained, and tested, and the results are evaluated.

Preparation for multi-class classification

The OMM-2022 dataset has just one feature (Class) which contains two distinct values (Malicious or Benign) and with this capability, we can just perform detection not category-wise or family-wise classification. To prepare it for categorization and family attribution, we used the first feature (Category) to prepare the dataset for multi-class classification. The malware categories, families, and hashes are extracted from each record and added to newly created features (Category, Family). A Python script is utilized to parse the dataset, extracting the first feature, parsing the text, and converting it into two new columns by eliminating the remaining string before deleting the original feature from the dataset. After completing this process, the new dataset includes two more characteristics (Category and Family) at the end. The feature “Class” is utilized to identify malware, while the newly developed features “Category” and “Family” are used to classify malware categories and attribute families.

Handling missing values

When data corrupts or is not recorded properly, missing or null values occur in the dataset, and handling these values is crucial. If the ML or DL algorithms are trained on such data, they produce incorrect results. We adopted a manual technique by keeping the Forward Fill/Backward Fill method in mind to handle the missing value of the attributes. After a comprehensive examination, a Python script is created that iterates on each row, and if it discovers a missing value in one of the columns, it first checks the target column value, followed by five backward and five forward values in the same columns. The OMM-2022 dataset has no missing values in the detection phase. However, after making adjustments and pre-processing for classification and family attribution, the dataset returned null values in the “Family” column. The null values in the Family field are replaced with “Benign” since the problem occurs only with “Benign” data. When we split the first column from “Category” into two columns, “Category” and ”Family”, the algorithm worked perfectly with malware records but not with benign records. The problem was that the “Category” column only had a “Benign” value for benign samples, with no family or hash value. The self-created dataset had no duplicate records since duplication was eliminated during the dataset development and cleaning process; however, the OMM-2022 dataset has 534 deleted duplicate entries.

Categorical encoding

Categorical encoding is the process of converting categorical values into numeric values and it must be converted before it feeds to ML or DL models. Typically, two approaches are used: one-hot encoding and simply substituting category data with numerical values. One-hot encoding is useful when the values are minimal and it is not appropriate for classification and family attribution tasks. The self-created and OMM-2022 datasets include three identical columns that must be transformed into numerical data. The first column is “Class” which has 2 unique values (Malicious and Benign) in both datasets and is converted into 0 and 1. In the self-created dataset, the second column, “Category”, contains five unique values (Benign, Trojan, Ransomware, Stealer, and RAT), whereas the OMM-2022 dataset has four unique values (Benign, Spyware, Ransomware, Trojan). The third column, “Family”, contains 27 unique entries in the self-created dataset and sixteen in the OMM-2022 dataset. All categorical values in both columns (Category and Family) are converted into machinable encoding.

Handling similarity and outliers

The OMM-2022 dataset is well-organized and has 2769.40 intra-class and 5821.51 inter-class similarity, which means there is no similarity issue. With a dataset size of 58,062 rows and 58 columns, computing pairwise distances for all data points can be computationally intensive and memory-consuming. However, given the calculation results, we can see that the intra-class similarity is lower than the inter-class similarity. This suggests that data points within the same class are, on average, more similar than data points from different classes, which is generally a desirable property for a well-behaved dataset. Given the dataset size, the results are pretty interpretable. They indicate that the data points are well-clustered within their respective classes and have more significant dissimilarity across different classes.

An observation that substantially differs from the other data points in a dataset is called an outlier. Measurement errors, natural variation, or exceptional cases can cause outliers. In machine learning, the presence of outliers can have a considerable impact on model performance. Statistical measurements like the mean and standard deviation can be distorted by outliers, which can result in erroneous depictions of the data distribution. This, in turn, can influence the behavior of certain ML algorithms sensitive to data distribution. Outliers may also disrupt the learning process by introducing noise and affecting the generalization ability of models. Addressing outliers is crucial for ensuring robust and reliable ML models. Depending on the specifications of the modeling job and the characteristics of the data, outlier detection, transformation, or removal may be used. Figure 2 shows outliers detected in the OMM-2022 dataset.

Handling outliers is a crucial step, and in this research, each feature is analyzed and handled separately. Sometimes having outliers does not indicate that there is some issue with the dataset. To handle outliers Z-sore technique is used and its mathematical formula for the calculation of the z-score is shown in Eq. (1). (1) Z=x−μσ

where Z presents the z-score, x is the individual data point, µis the mean of the dataset, and σ is the standard deviation of the dataset. For outlier detection using z-scores, a standard threshold is set (e.g., Z > 3orZ <  − 3), and data points beyond this threshold are considered outliers.

Data balancing

Balanced data is the ideal situation for ML and DL-related tasks and original datasets have no balancing issue, however, after removing duplicate entries it makes the OMM-2022 dataset a little imbalanced. Duplicate entries in malicious samples are removed. In the classification phase, datasets become imbalanced because the “Category” column has more than two unique values compared to benign samples, and the ratio becomes imbalanced. The dataset is balanced by first calculating the number of records category-wise, getting the average range of all categories, then randomly selecting the indices, dropping the selected benign records, concatenating the remaining benign and malicious records, shuffling the final DataFrame to mix benign and malicious records and then saving into new CSV file. This manual technique decreased the OMM-2022 dataset from 58,062 to 38,810. All values inside the “Family” column are converted into 16 unique entries (Benign, Transponder, Gator, Shade, Scar, Refroso, CWS, 180solutions, Ako, Conti, Emotet, Maze, Zeus, Pysa, Reconyc, and TIBS) against concerned category using a Python script and the same technique is used to balance the dataset. This technique decreased the OMM-2022 dataset from 39,318 to 30,816. Figure 3 depicts the picture of datasets before and after applying the manual technique.

Figure 2 Outliers detected in the OMM-2022 dataset.

Figure 3 The OMM-2022 dataset families before and after applying the balancing technique.

SMOTE (Synthetic Minority Over-sampling Technique) is a technique that increases synthetically the number of samples in a dataset in a balanced way. When there are many classes in a dataset that do not have an equal sample ratio, then an imbalanced data issue occurs. To handle and overcome this issue, the SMOTE technique produces synthetic data to make minority and majority classes of equal size. If the ratio of minority or majority classes is high, it is oversampled instead of undersampled to make equal-sized classes. The SMOTE selects examples closely related to the feature space and draws a new sample at a point along that line. Specifically, SMOTE generates synthetic samples for the minority class (or classes). For each sample in the minority class, the algorithm finds its k nearest neighbors (k is typically 5). Depending on the amount of over-sampling required, one or more of these k nearest neighbors are selected to create the synthetic samples. A synthetic sample is then created by choosing one of the k nearest neighbors and interpolating a new sample at a randomly selected point between the feature space of the minority sample and its selected nearest neighbor. In our multi-class classification approach, the SMOTE plays an essential role when datasets are prepared for the classification and family attribution tasks. After applying the SMOTE self-created dataset is increased from 15,638 to 23,810, and the OMM-2022 dataset from 38,810 to 40,016 in the classification phase. In the family attribution phase, the self-created dataset is increased from 11,426 to 14,850, and the OMM-2022 dataset from 30,816 to 38,544.

Feature selection

In this research, correlation and mutual information feature importance techniques are implemented on both datasets to check feature relations. After analyzing each feature, its importance concerning malware analysis, the values it contains, and its relation with the target’s essential features are selected and tested. If a feature has the same values in all benign and malware samples, then it is removed. A total of 50 53 features are selected from the OMM-2022 dataset, and 53 components are selected from the self-created dataset.

Normalization

Normalization is also called feature scaling, in which the values of features are normalized within a fixed range. The datasets used in this research have numeric values but some columns have high range and some with shallow range values. To increase ML and DL models’ efficiency, features are scaled from a high numeric value range to a fixed range from 0 to 1. The StandardScaler technique is employed to scale the data to a fixed range. The StandardScaler normalizes the data making each of the features have a mean = 0 while the standard deviation = 1.

Implementation of deep learning

The proposed model GN-BiLSTM is a combination of normalization layer GN and the DL model BiLSTM. The BiLSTM is the extension of the LSTM architecture, which captures information from past and future time steps in a sequence. This bidirectional working process helps the model to improve the understanding of context in sequential data. These models are mostly used in natural language processing (NLP) and in the application that captures contextual information essentially (Dang, Di Troia & Stamp, 2021). The BiLSTM network is composed of a forward LSTM hidden layer and a backward LSTM hidden layer in which data is processed in both (forward and backward) directions. The BiLSTM is differentiated from LSTM based on the backward direction which is only used in the BiLSTM, but not in the LSTM. The backward direction helps the model to capture hidden patterns of features of the data that are ignored by LSTM normally (Schuster & Paliwal, 1997). This bidirectional functionality helps and enables to capturing of more sophisticated tasks and more detail (Xiao et al., 2019). The architecture of the BiLSTM network can be seen in Fig. 4.

Figure 4 Structure of the BiLSTM.

Here, Hb is the forward layer and the backward layer is Hf while yt is the output sequence used in updating the work. It applied to update backward from ‘t’ to ‘1’ and forward from ‘1’ to ‘t’ step by step. The Eqs. (2), (3) and (4) are the mathematical expressions of the BiLSTM model. (2) Hf=σw1ytt+w2Hf−1+bLf

(3) Hb=σw2yit+w5Hb−1+bLb

(4) y=σw4Hf+w6H+bt

where Hf, Hb, and yt are forward, backward pass, and final output layers, respectively. w is the weights coefficient, and bLf, bLb, and bt are the biases (Shan et al., 2021). Group normalization (GN) is a normalization technique in which channels are divided into groups and features are normalized within each group. It does not modify the batch dimension because it computes independently of the batch size. The GN is defined in Eqs. (5), (6) and (7) if the group size is 1. (5) μi=1m∑keSixk

(6) σi2=1m∑keSinxk−μi2

(7) xi ˆ=xi−μiσi2+ϵ.

Here x is the computed feature by a layer, and i is an index. A Group Norm layer computes µand σ in a set Si and is defined in Eq. (8). (8) Si=kN=iN,kCC|G=ICC|G.

The number of groups is represented by the pre-defined hyper-parameter G, which by default equals 32. C|G indicates how many channels there are in each group. If each group of channels is maintained in a sequential sequence along the C axis, the last word indicates that the indexes i and k are in the same group of channels (Singla, Duhan & Saroha, 2022). The basic idea of this work is to improve ransomware detection, classification, and family attribution accuracy using Group group-normalized BiLSTM network. Figure 5 shows the structure of the GN-BiLSTM model.

Figure 5 Structure of the GN-BiLSTM.

Input sequence x = (x1, x2, x3, ………xt) where t is the sequence length. Hidden states ht forward and hr backward for each timestamp i.GN is the number of groups, γ, and β are learnable parameters. The Group Normalization operation is applied to each direction d of the BiLSTM for a given hidden state htd.

Equation (9) represents a reshaping operation that transforms the hidden state htd based on the batch size, group size, and the hidden state dimensions within each group. This operation organizes the hidden state into a new shape defined by BatchSize, the number of groups G, the size of each group (GroupSize), and the GroupedHiddenSize. (9) Reshape=htdBatchSize,G,GroupSize,GroupedHiddenSize

where BatchSize refers to the number of samples in each mini-batch, G represents the number of groups the hidden state is divided into, GroupSize is the size of each group, and GroupedHiddenSize is the size of the hidden state for each group. Equation (10) calculates the mean of the GroupedHiddenState across the batch and group dimensions. (10) Mean=MeanGroupedHiddenState,axis=BatchSize,GroupedSize

where GroupedHiddenState is the reshaped hidden state after grouping, the mean is computed along the dimensions defined by BatchSize and GroupedSize. Equation (11) computes the variance of the GroupedHiddenState, again across the BatchSize and GroupSize dimensions. (11) Variance=VarianceGroupedHiddenState,axis=BatchSize,GroupSize

where the variance calculation captures the spread of the grouped hidden states across the specified axes. Equation (12) normalizes the hidden state by subtracting the computed mean and dividing by the square root of the variance plus a small constant ϵ to prevent division by zero. (12) Normalizationhtd=htd−meanvariance+ϵ

where Mean and Variance are the mean and variance computed earlier, ϵ is a small constant added for numerical stability. Finally, Eq. (13) reshapes the normalized hidden state ytd back to its original shape of (BatchSize, HiddenSize) after the normalization process. (13) Reshapeytd,BatchSize,HiddenSizevariance+ϵ

where the normalized hidden state ytd is reshaped back to its original dimensions, and the scaling factor Variance+ϵ is applied to maintain numerical stability. These equations outline the steps for performing group normalization, where hidden states are normalized within groups for better stability and training performance.

To avoid bias, five other state-of-the-art DL models (CNN, MLP, LSTM, CNN-LSTM, and CNN-BiLSTM) are also implemented on both datasets. The CNN is ANN with many layers for image processing, detection, classification, and time series data, and is mainly developed for structured grid data processing like photos and videos (Ganfure et al., 2023). The convolutional layers use several kernels, also called learnable filters, to analyze the given input parameters (Yazdinejad et al., 2020). These filters move over the provided input images and analyze them to identify different elements like textures, edges, and patterns. Two important standard methods max-pooling and average-pooling minimize computational cost and complexity without changing translation invariance. After multiple convolutional and pooling layers, one or more fully connected layers are integrated by CNNs. These layers perform feature extraction and make predictions on the extracted features by connecting them to a classifier (Yazdinejad et al., 2020).

MLPs are more versatile, and useful and can handle a wider range of data types than CNNs because CNNs are focused on processing grid-like data (Hwang et al., 2020). MLP consists of three layers, an input layer, an output layer, and a hidden layer. Neurons apply neural activation functions on inputs in the hidden and output layers to achieve non-linearity (Dener, Ok & Orman, 2022). The hybrid model CNN-LSTM, a combination of CNN and LSTM, is widely used in the processing of sequential data, and due to its hybrid functionality, researchers have widely used it in malware detection and classification tasks. The convolutional layers apply kernels or learnable filters to the input data to capture edges, textures, spatial patterns, and other features related to the task. After the convolutional layers, one or more LSTM layers are used. These layers are responsible for capturing sequential dependencies within the spatial features extracted by the CNN. LSTM layers have memory cells that maintain information about past inputs, which allows them to model and understand the temporal relationships within the data. After the LSTM layers, fully connected layers (dense layers) can be added for further processing of the extracted features (Shafin, Karmakar & Mareels, 2023). CNN-Bi-LSTM is a neural network architecture that combines the strengths of both CNNs and BiLSTMs. The primary area where BiLSTM differs from LSTM is that it uses two hidden layers to process data in two directions. BiLSTM is widely used in the processing of natural language and other classification tasks (Aslan, Ozkan-Okay & Gupta, 2021).

Performance evaluation

Performance evaluation of the ML and DL models is an important task and to estimate the performance, the confusion metrics are considered the first choice to evaluate the results. Confusion metrics clarify the prediction of the model by measuring true positive (TP), true negative (TN), false positive (FP), and false negative (FN). Evaluation of DL-based model performance is to assess how well the model performed on a given task. Evaluation metrics vary depending on the type of task or problem being addressed by the DL models.

Accuracy

The first factor used to measure a model’s performance is to check its accuracy. It is reviewed and verified by watching a DL model’s accuracy and loss in each epoch and calculating average or mean accuracy at the end to predict the model’s accuracy. Equation (14) is the representation of accuracy. (14) Accuracy=TP+TNTP+FP+TN+FN

Precision

The second factor used to measure performance is precision. Precision is assessed by measuring the ratio of correctly identified positives by model and the total number of identified positives. Precision is presented in Eq. (15). (15) Precision=TPTP+FP.

Recall

The third important factor is recall. It is also called sensitivity, and it represents the ratio of linked instances retrieved to the overall number of retrieved instances. It is represented in Eq. (16). (16) Recall=TPTP+FN.

F-1 Score

The fourth important factor is considering the F1-score which is measured by considering both precision and recall. It is assumed to be the average weight of all values and is presented in Eq. (17). (17) F1−Score=2×Precision×RecallPrecision+Recall.

Results

This section provides a detailed performance analysis of implemented DL models. To assess their capabilities, three different detection tasks are conducted. The first one is malware detection, the second one is category-wise classification, and the third one is family attribution. Moreover, a comparison analysis is performed with the existing literature on the same dataset to validate the performance of the proposed approach.

The experiments are performed on an HP Z230 Workstation Core(TM) i7-4790 CPU @ 3.60 GHz (8 CPUs), equipped with a Windows 10 Profession 64-bit operating system with 16 Giga Byte Random Acess Memory (RAM). Additionally, the Google Colab platform is utilized by configuring Cloud Tensor Processing Units (TPUs) and Graphical Processing Units (GPUs) to check the computational performance. To implement practical work, popular software Anaconda 2.4.2 with Jupyter Notebook 6.4.12 and important libraries like Sklearn, TenserFlow, Numpy, and Pandas for ML and DL tasks are utilized. The dataset is divided into 80% for training and 20% for testing. K-Fold cross-validation is applied to ensure training and testing are performed on all classes. Table 2 describes the basic parameters set for all the DL models in training and testing. To avoid bias, the DL models are constructed with 32 batch size, 0.001 learning rate,, Adam optimizer for model optimization, and categorical cross-entropy function to track the loss and handle multi-class problems.

Table 2 Parameter detail of the implemented models.

Parameters	OMM-2022 dataset	Self-created dataset	
Batch size	32	32	
Epochs	20,30,50,100,150	20,30,50,100,150	
Learning rate	0.001	0.001	
Loss function	Categorical cross entropy	Categorical cross entropy	
Optimization algorithm	Adam	Adam	
Normalization	Standard	Standard	
Randomization	42	42	
Number of classes	50,51,52	46,47,48	
Cross-validation	K-Fold	K-Fold	
Number of splits	5	5	

Malware detection

At first, we experimented with malware detection by training DL models on both datasets using 20, 30, 50, 100, and 150 Epochs. All models are evaluated based on the average accuracy of the epoch, training time, and loss. The CNN achieved 99.99% accuracy with 0.0011 loss, 41 min for training on the OMM-2022 dataset, and 97.40% accuracy with 0.0828 loss, 10 min for training on the self-created dataset. The MLP achieved 99.99% accuracy with 0.0007 loss, 18 min for training on the OMM-2022 dataset, and 72.92% accuracy with 0.7899 loss, 5 min for training on the self-created dataset. The LSTM achieved 99.99% accuracy with 0.0010 loss, 74 min for training on the OMM-2022 dataset, and 74.20% accuracy with 0.4916 loss, 29 min for training on the self-created dataset. The CNN-LSTM achieved 99.97% accuracy with 0.0011 loss, 61 min for training on the OMM-2022 dataset, and 99.09% accuracy with 0.0263 loss, 39 min for training on the self-created dataset. The CNN-Bi-LSTM achieved 99.97% accuracy with 0.0012 loss, 95 min for training on the OMM-2022 dataset, and 99.15% accuracy with 0.0277 loss, 39 min for training on the self-created dataset. Our proposed model the GN-BiLSTM achieved 99.99% accuracy on the OMM-2022 dataset and 99.20% on the self-created dataset. Tables 3 and 4 provide the average accuracy, loss, and training time of models on both datasets in the malware detection phase.

Table 3 Average training time, loss, and accuracy on the OMM-2022 dataset.

Model	Average time (Minutes)	Average loss	Average accuracy	
CNN	41	0.0011	99.99%	
MLP	18	0.0007	99.99%	
LSTM	74	0.0010	99.99%	
CNN-LSTM	61	0.0011	99.97%	
CNN-BiLSTM	95	0.0012	99.97%	
GN-BiLSTM	131	0.0002	99.99%	

Table 4 Average training time, loss, and accuracy on the self-created dataset.

Model	Average time (Minutes)	Average loss	Average accuracy	
CNN	10	0.0828	97.40%	
MLP	5	0.7899	72.92%	
LSTM	29	0.4916	74.20%	
CNN-LSTM	39	0.0263	99.09%	
CNN-BiLSTM	68	0.0277	99.15%	
GN-BiLSTM	38	0.0186	99.20%	

The performance of the models on the OMM-2022 dataset can be seen in Fig. 6. All models performed well on the OMM-2022 dataset, with accuracy rates approaching 99.99%. The GN-BiLSTM, CNN, and MLP models all scored an excellent 99.99% accuracy, while the CNN-LSTM and CNN-BiLSTM models achieved a slightly lower 99.97%. Although this difference in accuracy appears little, it demonstrates the GN-BiLSTM model’s durability, since it retains greater accuracy despite its more complicated design. The incorporation of bidirectional LSTMs in the GN-BiLSTM architecture allows it to better capture temporal relationships in data, resulting in higher performance while processing obfuscated malware samples.

Figure 6 Models accuracy and loss on the OMM-2022 dataset.

The average loss metric emphasizes the efficacy of the GN-BiLSTM model. The GN-BiLSTM performs substantially better than all other models, with an average loss of 0.0002. The CNN, LSTM, and CNN-LSTM models all have larger loss values ranging from 0.0010 to 0.0012, showing that while these models may still attain high accuracy, their predictions are less reliable than the GN-BiLSTM. The reduced loss in the GN-BiLSTM indicates that the model’s predictions are more exact and can handle edge circumstances better than other designs. This minimal loss, paired with excellent accuracy, demonstrates the GN-BiLSTM model’s capacity to generalize successfully, making it more resistant to new and disguised ransomware variants.

Despite the longer training period, the GN-BiLSTM’s improved performance metrics support its application in cases requiring high accuracy and generalization. In actual applications, slight increases in accuracy can be critical for identifying sophisticated or obfuscated malware that other models may miss. Furthermore, by accelerating hardware or distributed computing, it may reduce the computational cost, making the GN-BiLSTM a more suitable model for large-scale or real-time applications. The ability of the GN-BiLSTM model to achieve low average loss and high accuracy suggests that it can better adapt to fluctuations in malware patterns, allowing for continuing scalability by reducing the need for regular retraining in dynamic contexts.

The performance of models on the self-created dataset can be seen in Fig. 7. The GN-BiLSTM model outperformed other models on the self-created dataset, with an accuracy of 99.20%. The CNN, MLP, and LSTM models had much lower accuracies (97.40%, 72.92%, and 74.20%, respectively). The CNN-LSTM and CNN-BiLSTM models both performed well, with accuracies of 99.09% and 99.15%, respectively. However, the GN-BiLSTM’s somewhat greater accuracy reflects its improved ability to detect and classify ransomware in the self-created dataset, which may contain varied and innovative ransomware samples. This illustrates that the GN-BiLSTM excels at detecting subtle characteristics in data, retaining excellent accuracy even when dealing with fresh or complicated samples.

Figure 7 Models accuracy and loss on the self-created dataset.

The GN-BiLSTM model not only has the best accuracy but also the lowest average loss (0.0186), suggesting extremely confident predictions. In comparison, the CNN model had a significantly larger average loss of 0.0828, while the MLP and LSTM models showed even greater losses of 0.7899 and 0.4916, respectively. The CNN-LSTM and CNN-BiLSTM models both have low losses (0.0263 and 0.0277), however, the GN-BiLSTM’s lower loss emphasizes its better performance. This lower loss indicates that the GN-BiLSTM can discriminate between benign and malicious samples with greater precision, which is critical for successful ransomware detection.

The GN-BiLSTM model proved its scalability and reliability by training and producing results on the self-created dataset. Despite the longer training time of the model compared to the CNN and CNN-LSTM, its high accuracy and small loss make it ideal for practical use. The accuracy it achieved and the loss it decreased indicate that it can easily and efficiently handle a wide range of ransomware families, including previously unknown ones. The mode’s exceptional ability to generalize from data and efficiency makes it a strong excellent candidate for real-world ransomware classification systems, especially where precise classification is required.

On the OMM-2022 dataset, which contains samples of 29,298 samples of obfuscated malware, the performance of the GN-BiLSTM is outstanding and it achieved higher accuracy 99.99% with 0.0002 minimal loss compared to other models in the same categories. Despite, its long training time of 131 min, its excellent accuracy with very low loss demonstrates its ability that it can detect and classify obfuscated malware accurately. The CNN, MLP, and LSTM model also achieved high accuracy; however, their losses are very high. The CNN-LSTM and CNN-BiLSTM models also performed well but they do not match the accuracy and loss that the GN-BiLSTM model achieved. This highlights the GN-BiLSTM model’s exceptional generalization abilities and tolerance to sophisticated malware.

The GN-BiLSTM model retained its high accuracy of 99.20% and lowest average loss of 0.0186 on the self-created dataset which includes ransomware samples from 11 families. The CNN-LSTM and CNN-BiLSTM also achieved high accuracy but a little lower than the GN-BiLSTM model; however, their loss is higher than the GN-BiLSTM. The training time of the GN-BiLSTM is 38 min, which is longer than the CNN, MLP, and CNN-LSTM but lower than the CN-BiLSTM, however, the accuracy and loss metrics of the GN-BiLSTM model justify the time. The balance of accuracy, loss, and training time of the GN-BiLSTM model highlights its ability to handle a diverse set of ransomware samples and its potential for real-world applications. The GN-BiLSTM model performed efficiently on both datasets and maintained consistency in obtaining excellent accuracy and low loss. Its ability to classify malware from benign with high accuracy and minimal loss, especially when dealing with obfuscated ransomware samples, makes it an ideal choice for practical applications of ransomware classification.

Malware category identification

Results from the detection phase are excellent, but just detecting malware samples without knowing their categories does not complete the defense. It is important to identify its correct category to prepare an exact defense against it. The OMM-2022 dataset contains four categories, whereas the self-created dataset has five. It is critical to understand the category of malware. In this classification phase, we trained models to identify categories of three classes of malware (ransomware, spyware, Trojan) from the OMM-2022 dataset and four classes (ransomware, RAT, trojan, stealer) from the self-created dataset other than benign class.

In the category-wise malware classification phase, the same epoch-wise parameters (20, 30, 50, 100, and 150) are used to train and test DL models on both datasets. Models are evaluated based on average accuracy, training time, and loss on each epoch. The CNN achieved 69.60% accuracy with 0.6602 loss, 21 min for training on the OMM-2022 dataset, and 89.68% accuracy with 0.3494 loss, 8 min for training on the self-created dataset. The MLP achieved 72.96% accuracy with 0.6051 loss, 5 min for training on the OMM-2022 dataset, and 91.74% accuracy with 0.2847 loss, 5 min for training on the self-created dataset. The LSTM achieved 74.70% accuracy with 0.5571 loss, 45 min for training on the OMM-2022 dataset, and 93.72% accuracy with 0.2267 loss, 16 min for training on the self-created dataset. The CNN-LSTM achieved 75.19% accuracy with 0.4925 loss, 28 min for training on the OMM-2022 dataset, and 95.45% accuracy with 0.1826 loss, 12 min for training on the self-created dataset. The CNN-Bi-LSTM achieved 76.09% accuracy with 0.5032 loss, 104 min for training on the OMM-2022 dataset, and 95.22% accuracy with 0.1092 loss, 30 min for training on the self-created dataset. The GN-Bi-LSTM achieved 85.45% accuracy with 0.3553 loss, 74 min for training on the OMM-2022 dataset, and 96.44% accuracy with 0.0655 loss, 33 min for training on the self-created dataset. Tables 5 and 6 provide the average accuracy, loss, and training time of all six models on both datasets in malware classification.

Table 5 Average training time, loss, and accuracy on the OMM-2022 dataset.

Model	Average time (Minutes)	Average loss	Average accuracy	
CNN	21	0.6602	69.60%	
MLP	5	0.6051	72.96%	
LSTM	45	0.5571	74.70%	
CNN-LSTM	29	0.4925	75.19%	
CNN-BiLSTM	104	0.5032	76.09%	
GN-BiLSTM	74	0.3553	85.48%	

Table 6 Average training time, loss, and accuracy on the self-created dataset.

Model	Average time (Minutes)	Average loss	Average accuracy	
CNN	8	0.3494	89.68%	
MLP	5	0.2847	91.74%	
LSTM	16	0.2267	93.72%	
CNN-LSTM	12	0.1826	95.45%	
CNN-BiLSTM	30	0.1092	95.22%	
GN-BiLSTM	33	0.0655	97.44%	

Figure 8 shows the accuracy and loss graphs of all six models in classifying malware categories on the OMM-2022 dataset. The GN-BiLSTM model achieved the highest accuracy of 85.48% in classifying malware families and this is a significant development over the CNN, MLP, and LSTM models which achieved 69.60%, 72.96%, and 74.70% accuracy respectively. The CNN-LSTM and CNN-BiLSTM models performed better than the CNN and MLP by achieving 75.19% and 76.09% accuracy respectively. However, outstanding accuracy and low loss from the GN-BiLSTM reveals its efficacy in identifying between various malware categories, and its greater ability to complete complicated classification tasks with better reliability when compared to its rivals.

Figure 8 Models accuracy and loss on the OMM-2022 dataset.

The GN-BiLSTM model achieved the lowest average loss of 0.3553 from the tested models, significantly dropping from the CNN, MLP, and LSTM models, which achieved 0.6602, 0.6051, and 0.5571 loss respectively. The CNN-LSTM and CNN-BiLSTM also achieved a low loss of 0.4925 and 0.5032 respectively, but higher than the GN-BiLSTM. The lower loss from the GN-BiLSTM indicates that it can make more accurate predictions, providing its efficiency in categorizing malware groups more precisely.

The GN-BiLSTM model’s average training time is 74 min longer than the CNN, MLP, and LSTM which took 21, 5, and 45 min respectively. The CNN-BiLSTM took 104 min in training which is longer than the GN-BiLSTM. Despite the GN-BiLSTM model taking a long training time, its higher accuracy and lower loss justify it. The trade-off between category-wise classification accuracy and training length reveals the GN-BiLSTM model’s robustness, meaning that additional training time provides much better results. The remarkable performance of the GN-BiLSTM model on the OMM-2022 dataset shows its resilience in identifying malware categories. Its balanced training time with high accuracy and low loss, make it a perfect candidate for practical use in real-world applications requiring precise malware classification. Although it requires longer training time than the other models, the large increase in accuracy and loss reduction demonstrate its usefulness.

Figure 9 provides the performance of all utilized models in classifying malware categories on the self-created dataset. The GN-BiLSTM achieved 97.44%, CNN 89.68%, MLP 91.74%, LSTM 93.72%, CNN-LSTM 95.45%, and CNN-BiLSTM 95.22% accuracy respectively. The excellent accuracy and low loss of the GN-BiLSTM model on the self-created dataset highlights its advanced capabilities in identifying malware categories, demonstrating its robustness and effectiveness in dealing with the various malware samples. The GN-BiLSTM model also had the lowest average training loss of 0.0655 compared to the CNN (0.3494), MLP (0.2847), and LSTM (0.2267), which demonstrates its ability to produce more confident predictions with less inaccuracy. The CNN-LSTM achieved 0.1826 and the CNN-BiLSTM 0.1092 losses, which are less than the other utilized models, however, the GN-BiLSTM model beat all rival models by reducing loss classification errors and its usefulness in malware classification.

Figure 9 Models accuracy and loss on the self-created dataset.

On the OMM-2022 dataset, which includes different malware samples, the GN-BiLSTM performed exceptionally well and achieved an average of 85.48% accuracy with 0.3553 loss in categorizing malware. Its high accuracy demonstrates its ability to identify between different malware types efficiently when compared to other utilized models. The CNN model achieved 69.60% accuracy, MLP achieved 72.96%, LSTM 74.70%, CNN-LSTM 75.19%, and CNN-BiLSTM 76.09% in identifying malware categories. Results indicate that the GN-BiLSTM model outperformed all rival models, despite the extended training period, its higher classification accuracy and minimal loss demonstrate its ability to handle sophisticated malware categorization jobs.

On the self-created dataset, which includes a more diversified range of malware samples, the GN-BiLSTM model beat all other utilized models by achieving 97.44% accuracy with 0.0655 loss. This highlights its outstanding ability to categorize malware types correctly across a broad dataset. Although, the CNN model has the shortest training time but achieved the lowest accuracy of 89.68% and the highest loss of 0.3494. The MLP achieved 91.74% accuracy and LSTM 93.72% but were outperformed by the GN-BiLSTM model’s performance. The CNN-LSTM and CNN-BiLSTM models outperformed the CNN and MLP with accuracy of 95.45% and 95.22% respectively, however, fell short of the GN-BiLSTM model’s performance in categorizing malware. Despite it requires more training time than the other models, the high accuracy and minimal loss of the GN-BiLSTM illustrate its usefulness and resilience in classifying malware categories.

The GN-BiLSTM model consistently outperformed in categorizing malware samples in both datasets and its excellent accuracy with minimal loss demonstrates its usefulness and reliability in discriminating between malware categories. While the GN-BiLSTM requires a longer training time, a considerable boost in the overall classification accuracy and loss reduction highlights the model’s robustness and applicability for practical applications. Its outstanding performance reveals its ability to handle more sophisticated and difficult classification tasks and makes it an important practical tool for accurate malware types.

Malware family attribution

It is considerable that each malware family may have different unique patterns, features, and obfuscation techniques applied to avoid detection. The primary goal of the research is to classify ransomware families correctly with high accuracy. There are a total of 13 classes in the OMM-2022 dataset and 27 classes in the self-created dataset. Therefore, evaluation of the proposed model’s performance in the classification of all individual classes would verify its ability to attribute malware families.

Using the same epoch-wise parameters utilized in detection and classification, the CNN achieved 55.47% accuracy with 1.1269 loss, 18 min for training on the OMM-2022 dataset, and 92.43% accuracy with 0.3034 loss, 10 min for training on the self-created dataset. The MLP achieved 58.98% accuracy with 1.0590, 10 min for training on the OMM-2022 dataset, and 92.10% accuracy with 0.3269 loss, 2 min for training on the self-created dataset. The LSTM achieved 63.35% accuracy with 0.9328 loss, 42 min for training on the OMM-2022 dataset, and 92.25% accuracy with 0.2978 loss, 9 min for training on the self-created dataset. The CNN-LSTM achieved 67.10% accuracy with 0.785 loss, 51 min for training on the OMM-2022 dataset, and 94.54% accuracy with 0.1654 loss, 22 min for training on the self-created dataset. The CNN-Bi-LSTM achieved 61.17% accuracy with 0.8301 loss, 70 min for training on the OMM-2022 dataset, and 93.54% accuracy with 0.1857 loss, 47 min for training time on the self-created dataset. GN-Bi-LSTM achieved 74.65% accuracy with 0.4898 loss, 54 min for training on the OMM-2022 dataset, and 96.23% accuracy with 0.0699 loss, 31 min for training on the self-created dataset. Tables 7 and 8 provide the average accuracy, loss, and training time of models on both datasets in the malware family attribution phase.

Table 7 Average training time, loss, and accuracy on the OMM-2022 dataset.

Model	Average time (Minutes)	Average loss	Average accuracy	
CNN	18	1.1269	55.47%	
MLP	10	1.0590	58.98%	
LSTM	42	0.9328	63.35%	
CNN-LSTM	51	0.7857	67.10%	
CNN-BiLSTM	70	0.8301	61.17%	
GN-BiLSTM	54	0.4898	74.65%	

Table 8 Average training time, loss, and accuracy on the self-created dataset.

Model	Average time (Minutes)	Average loss	Average accuracy	
CNN	10	0.3034	92.43%	
MLP	2	0.3269	92.10%	
LSTM	9	0.2978	92.25%	
CNN-LSTM	22	0.1654	94.54%	
CNN-BiLSTM	47	0.1857	93.54%	
GN-BiLSTM	31	0.0699	96.23%	

Figure 10 shows the accuracy and loss graphs of all six models in attributing malware families. The GN-BiLSTM model has the greatest accuracy of 74.65% on the OMM-2022 dataset in classifying malware families. This is much higher than other models, which attained accuracies of 55.47%, 58.98%, and 63.35%, respectively. The CNN-LSTM and CNN-BiLSTM models also performed well, with accuracies of 67.10% and 61.17%, respectively. The considerable accuracy gain achieved with GN-BiLSTM demonstrates its effectiveness in distinguishing between distinct malware families, exhibiting advanced skills in dealing with complex classification situations.

Figure 10 Models accuracy and loss on the OMM-2022 dataset.

The GN-BiLSTM also has the lowest average loss of 0.4898, which is much lower than the CNN, MLP, and LSTM which produced 1.1269, 1.0590, and 0.9328 loss. The CNN-LSTM with 0.7857 and CNN-BiLSTM with 0.8301 losses show that these models incorrectly predicted a large portion of malware samples. The lowest loss of the GN-BiLSTM model indicates that it can produce more accurate results with greater accuracy and reduce classification errors in malware family identification. The GN-BiLSTM model took 54 min average training time, which is longer than the CNN, MLP, and LSTM models, which took 18, 10, and 42 min respectively. However, it took less time than the CNN-LSTM and CN-BiLSTM which took 51 and 70 min respectively in training. The GN-BiLSTM model, though took longer training time than the rival models, but its higher accuracy and lower loss justify its extra efforts in identifying malware families.

On the OMM-2022 dataset, the GN-BiLSTM model’s outstanding performance in categorizing malware families indicates its reliability and efficiency. Its high accuracy and low loss while identifying malware families make it an excellent choice for practical applications where exact malware family identification is required. Despite its longer training time, the increase in accuracy and decrease in loss demonstrate its aptitude for sophisticated malware family classification jobs. Its ability to differentiate from malware families indicates its potential to be implemented in real-world applications for reliable and effective classification and threat detection tasks.

Figure 11 shows the accuracy and loss of all models in categorizing malware families from the self-created dataset. The GN-BiLSTM model performed excellently and achieved 96.23% highest accuracy in identifying malware families. The CNN achieved 92.43%, MLP 92.10%, and LSTM 92.25% accuracy while the CNN-LSTM and CNN-BiLSTM outperformed all these three models by achieving 94.54% and 93.54% accuracy respectively. However, The GN-BiLSTM model’s high accuracy demonstrates its extraordinary performance in accurately capturing malware families.

Figure 11 Models accuracy and loss on the self-created dataset.

The GN-BiLSTM also outperformed other utilized models with the lowest average loss of 0.0699 in family-wise identifying malware. The CNN model had 0.3034, MLP 0.3269, LSTM 0.2978, CNN-LSTM 0.1654, and CNN-BiLSTM 0.1857 losses in attributing malware families. The efficient decrease in loss and increase in accuracy values by the GN-BiLSTM model highlights its trustworthiness and usefulness in reducing classification errors and demonstrates its ability to make more exact predictions.

The average training time taken by the GN-BiLSTM is 31 min in categorizing malware families, which is more than the MLP which took 2 min, LSTM 9 min, and CNN-LSTM 22 min; however, these times were shorter than CNN-BiLSTM, which took 47 min average training time in malware families classification. Despite its longer training time compared to other models, the GN-BiLSTM achieved the highest accuracy with reduced loss which justifies its training time. The trade-off between category-wise classification performance and training time demonstrates its capabilities to maintain high accuracy while reducing loss and acceptable training time.

The performance of the GN-BiLSTM model on the self-created dataset reveals its strong capabilities for classifying malware families. Its high accuracy with low loss and balanced training time makes it an ideal choice for real-world applications that need exact malware family classification. Its ability to provide higher results even with a diverse set of malware samples demonstrates its usefulness in real-world applications, implying that it is well-suited for complex classification tasks.

On the OMM-2022 dataset, the GN-BiLSTM model performed exceptionally well while classifying malware families achieving an accuracy of 74.65% with 0.4898 average loss. The GN-BiLSTM outperformed the CNN, MLP, and LSTM which achieved 55.47%, 58.98%, and 63.35% accuracies respectively and showcased its better ability to effectively identify correct malware families with minimum errors. Although the GN-BiLSTM took a longer training time of 54 min compared to the CNN which took 10 min and MLP which took 18 min, its performance increase in accuracy and decrease in loss outweighed its extra training time. Its ability to maintain high accuracy and reduce loss demonstrates its usefulness in sophisticated categorization problems. Its strong performance throughout the dataset demonstrates its ability to discriminate between different malware families, making it a useful method for real-world ransomware classification tasks.

One of the important aspects in the evaluation of any ransomware classification system is its ability to respond to new and developing ransomware threats. The adaptability of the GN-BiLSTM-based method is largely dependent on its ability to learn patterns from existing data. While the GN-BiLSTM demonstrates robust performance against known ransomware families, the rise of novel types of attacks poses a significant challenge. It is intended to capture complex connections and sequential connections in ransomware activity because of its bidirectional learning and graph-based feature representation. Even in cases when the training data does not contain the particular ransomware kind, these features allow the model to potentially identify unusual behavior that deviates from recognized attack patterns. The effectiveness of this classification, however, depends on how closely the new attack resembles known ransomware patterns.

In both the OMM-2022 and self-created datasets, the GN-BiLSTM consistently outperformed other models in family-wise malware classification, with greater accuracy and lower loss rates. The model’s excellent performance, despite longer training durations than some other models, demonstrates its durability and efficacy in differentiating malware families. The GN-BiLSTM’s dependability and effectiveness in dealing with a wide range of malware samples highlight its significance as a potent tool for practical malware detection and classification. This thorough examination of the GN-BiLSTM’s performance across several datasets confirms its advanced capabilities and applicability for real-world applications in malware family classification.

Discussion

As per the reviewed literature, this study is not the first contribution to identifying malware families. However, implementation and verification of the proposed model on real-world samples have not been addressed. We showed the evaluated results of all 6 models in detection, classification, and family attribution in Tables 9 and 10 on the OMM-2022 dataset. Our proposed model achieved the highest accuracy of 99.99%, 85.48%, 74.65%, and more minor training loss. Figure 12 presents the evaluated results graphically.

Table 9 Average accuracy in all three phases on the OMM-2022 dataset.

Model	Detection	Classification	Family attribution	
CNN	99.99%	69.60%	55.47%	
MLP	99.99%	72.96%	58.98%	
LSTM	99.99%	74.70%	63.35%	
CNN-LSTM	99.97%	75.19%	67.10%	
CNN-BiLSTM	99.97%	76.09%	61.17%	
GN-BiLSTM	99.99%	85.48%	74.65%	

Table 10 Average loss in all three phases on the OMM-2022 dataset.

Model	Detection	Classification	Family attribution	
CNN	0.0011	0.6602	1.1269	
MLP	0.0007	0.6051	1.0590	
LSTM	0.0010	0.5571	0.9328	
CNN-LSTM	0.0011	0.4925	0.7857	
CNN-BiLSTM	0.0012	0.5032	0.8301	
GN-BiLSTM	0.0002	0.3553	0.4896	

Figure 12 Models average accuracy and loss in all three phases on the OMM-2022 dataset.

On the OMM-2022 dataset, the GN-BiLSTM model outperforms in all three evaluation phases: detection, category-wise classification, and family-wise classification. On the OMM-2022 dataset, the GN-BiLSTM model performs exceptionally well in terms of average loss throughout all three assessment stages (detection, category-wise classification, and family-wise classification).

The GN-BiLSTM achieved 99.99% detection accuracy, comparable to CNN and MLP models. However, its performance is distinguished by its constant efficacy across all stages, indicating its capability to classify malware with few false positives or negatives. The GN-BiLSTM had the lowest average loss of 0.0002 in the malware detection phase, which is much lower than the other rival models like CNN which had 0.0011, MLP 0.0007, CNN-LSTM 0.0011 and CNN-BiLSTM 0.0012 losses. The high accuracy and low loss from the GN-BiLSTM model demonstrate its remarkable accuracy in identifying malware from benign samples, highlighting its detection accuracy and low risk of misclassification.

In classifying malware categories, the GN-BiLSTM beat all other models in achieving high accuracy and low loss. The CNN achieved 69.60%, MLP 72.96%, LSTM 74.70%, CNN-LSTM 75.19%, and CNN-BiLSTM 76.09% accuracies which are lower than the proposed model’s performance. This high accuracy from the GN-BiLSTM reflects its ability to classify different malware categories, its efficiency in dealing with complex classification tasks, and its ability to give more accurate categorizations than earlier models. The GN-BiLSTM model has an average loss of 0.3553, which is lower than the CNN which had 0.6602, ML 0.6051, LSTM 0.5571, CNN-LSTM 0.4925, and CNN-BiLSTM 0.5032 losses. Its decreased loss indicates that it excels at classifying malware categories with fewer errors, providing its improved ability to handle multifaceted classification tasks and its use in giving more accurate results.

The GN-BiLSTM beat all other models in classifying malware families by achieving high accuracy with minimal loss. The CNN achieved 55.47%, MLP 58.98%, LSTM 63.35%, CNN-LSTM 67.10%, and CNN-BiLSTM 61.17% accuracies, which are lower than the proposed model’s accuracy. It had also the lowest average loss of 0.4896 compared to other models like CNN had 1.1269, MLP 1.0590, LSTM 0.9328, CNN-LSTM 0.7857, and CNN-BiLSTM 0.8301 loss. This highlights the proposed model’s ability to provide more accurate results in classifying malware families even for complex and different malware samples.

The performance of the GN-BiLSTM model is outstanding in all three phases, including the classification of malware categories and families. Its continuously maintained high performance on these critical assessment parameters discloses its strength and efficiency. Despite the other models having exceptional detection capabilities and results, the GN-BiLSTM differentiates itself by offering greater classification accuracy, indicating its greater capability to manage a wide range of malware concerns. Its consistently low average loss in all three phases highlights its exceptional performance, reliability, and consistency in eliminating errors and generating more accurate results. Although other models performed well, the GN-BiLSTM model’s loss metrics indicate its improved ability to tackle a variety of complex malware classification tasks. Its extensive loss analysis shows that it is an exceptional model for malware categorization and family-wise identification, showing its efficiency and practical use in real-world applications. The proposed model stands out in malware analysis for its high accuracy rate in not only detection but also classification of categories and families. Its balanced performance through all three stages increases its trustworthiness and practicality for complete malware detection and classification tasks.

Tables 11 and 12 provide the evaluated results of all six trained models in all three phases (detection, category-wise and family-wise classification) on the self-created dataset. The proposed model achieved the highest accuracy score of 99.20% in detecting malware, 97.44% in classifying its category, and 96.23% in attributing its family. Figure 13 is the graphical representation of evaluated models.

Table 11 Average accuracy in all three phases on the self-created dataset.

Model	Detection	Classification	Family attribution	
CNN	97.40%	89.68%	92.43%	
MLP	72.92%	91.74%	92.10%	
LSTM	74.20%	93.72%	92.25%	
CNN-LSTM	99.09%	95.45%	94.54%	
CNN-BiLSTM	99.15%	95.22%	93.54%	
GN-BiLSTM	99.20%	97.44%	96.23%	

Table 12 Average loss in all three phases on the self-created dataset.

Model	Detection	Classification	Family attribution	
CNN	0.0828	0.3494	0.3034	
MLP	0.7899	0.2847	0.3269	
LSTM	0.4916	0.2267	0.2978	
CNN-LSTM	0.7899	0.1826	0.1654	
CNN-BiLSTM	0.4916	0.1092	0.1857	
GN-BiLSTM	0.0186	0.0655	0.0699	

Figure 13 Models average accuracy and loss in all three phases on the self-created dataset.

The GN-BiLSTM model performed excellently on the self-created dataset in all three phases of detection and classification. In the malware detection phase, the GN-BiLSTM achieved the highest accuracy of 99.20% by outperforming the CNN (97.40%), MLP (72.92%), LSTM (74.20%), CNN-LSTM (99.09%), and CNN-BiLSTM (99.15%). Furthermore, it had the lowest average loss of 0.0186 when compared to the CNN, MLP, LSTM, CNN-LSTM, and CNN-BiLSTM, which had 0.0828, 0.7899, 0.4916, 0.7899, and 04,916 losses receptively. This provides the outstanding capability of the GN-BiLSTM to classify malware properly while minimizing loss, which can be seen by its high accuracy and low risk of false positives and false negatives in real-world environments.

In classifying malware categories, the GN-BiLSTM model has the highest average accuracy rate of 97.44% among all models. It outperformed CNN (89.68%), MLP (91.74%), LSTM (93.72%), CNN-LSTM (95.45%), and CNN-BiLSTM (95.22%). It also has the smallest average loss of 0.0655, beating the CNN, MLP, LSTM, CNN-LSTM, and CNN-BiLSTM, which have 0.3494, 0.2847, 0.2267, 0.1826, and 0.1092 average loss. This highlights that it excels at classifying malware into specific categories and families with high accuracy and low errors, providing its advanced classification capabilities.

In classifying malware families, the GN-BiLSTM model achieved the highest accuracy of 96.23%, topping CNN (92.43%), MLP (92.10%), LSTM (92.25%), CNN-LSTM (94.54%), and CNN-BiLSTM (93.54%). It also has the lowest average loss of 0.0699, which is much lesser than the CNN, MLP, LSTM, CNN-LSTM, and CNN-BiLSTM, which had 0.3034, 0.3269, 0.2978, 0.1654, and 0.1857 loss. This study highlights the GN-BiLSTM’s amazing ability to detect malware families with little classification error.

Several tests are performed in this work to validate the proposed GN-BiLSTM-based ransomware detection system. The results demonstrated how well the algorithm performed overall in identifying and classifying malware. However, a more detailed cause analysis provides a more complete understanding of the underlying variables that influenced these outcomes. The GN-BiLSTM model’s superior performance can be ascribed to its ability to handle temporal and spatial links in the dataset. The model can detect sequential patterns in ransomware activity due to the bidirectional layers of the LSTM, which more traditional machine learning methods often miss. This improves its accuracy of multi-class classification by identifying tiny changes in attack signatures.

The quantity and quality of available data for certain classes might clarify some of the performance changes across different ransomware categories, even if the results showed promising detection rates. The model’s capability to completely generalize may be vulnerable to sparse or limited data for particular ransomware families. Ensuing versions of this research might reap advantages from expanding the dataset to incorporate a wider range of ransomware variations and highlighting enhanced feature engineering to capture additional characteristic attributes of every category. Furthermore, even though feature selection helped to reduce the dimensionality of the data, it is possible that certain traits that were vital for classifying more advanced ransomware attacks were unintentionally left out. Increasing feature variety and investigating alternative feature selection techniques may help to reinforce the model’s flexibility. In conclusion, the suggested GN-BiLSTM system’s capacity to recognize intricate patterns and correlations in the data allowed it to perform well in ransomware detection. On the other hand, by resolving data restrictions and investigating new ways to further improve detection accuracy, future work can continue to improve the model.

Comparison

The comparison mainly focused on existing works on the OMM-2022 dataset to avoid bias. Table 13 shows the results of the five various related works on the target dataset. The criteria for comparison are overall detection, classification, and family attribution accuracy of the technique, learning approach, implementation of cross-validation, and verifying technique by applying on some other real-world dataset. Based on these criteria, the highest results of each work are chosen. These five related works are compared to the proposed model, Detection, Classification, and Family Attribution of Obfuscated Malware using GN-BiLSTM.

Table 13 Comparison of GN-BiLSTM with other state-of-the art work on same dataset.

Reference	Technique	Validation	Malware	Category	Family	
Roy et al. (2023)	ML: Stacked ensemble	4-K cross-validation	99.98%	85.04%	70.29%	
Abualhaj et al. (2024)	ML: KNN	–	99.97%	82.21%	66.93%	
Mezina & Burget (2022)	DL: Dilated CNN	–	99%	–	83%	
Roy & Chen (2021)	DL: TF-IDF	–	99.87%	–	96.5%	
Shafin et al. (2023)	–	–	99.96%	84.56%	72.60%	
Proposed GN-BiLSTM	DL: Feature engineering & SMOTE	K-Fold Cross-validation	99.99%	85.48%	74.68%	

Roy et al. (2023) used an ML-based approach with a stacked ensemble learning scheme and applied 4-k cross-validation. They achieved inspiring results in all three phases (detection, classification, and family attribution). However, they did not apply the technique to other datasets to verify its real-world implementation. In the second selected work (Abualhaj et al., 2024), the author also used an ML-based approach with the Pearson correlation coefficient technique for feature selection and improving distance metric parameters of the KNN. K-Fold cross-validation is applied, and the detection and classification results are impressive. However, the family attribution results are low when compared to other works. In Mezina & Burget (2022), the DL-based dilated CNN method is proposed to detect obfuscated malware and classify its family. However, they only performed malware detection and family attribution but not malware category classification. Cross-validation and implementation of the model are not checked with some other datasets to verify its practical application. In Roy & Chen (2021), a DL-based approach using the TF-IDF technique for binary and multi-class classification is used. They achieved very high scores in multi-class classification, however, they did not apply any cross-validation or verify their technique by applying it to some other latest real-world datasets. In Shafin, Karmakar & Mareels (2023), the DL-based approach is proposed and achieves impressive results in all phases. However, they did not implement any novel technique or cross-validation. Verification of the implemented model is also not performed on some of the latest datasets.

When the GN-BiLSTM is compared to other cutting-edge algorithms for malware detection, our model outperforms them all. The GN-BiLSTM has an exceptional detection accuracy of 99.99%, exceeding (Roy et al., 2023) at 99.98%, (Abualhaj et al., 2024) at 99.97%, and Shafin, Karmakar & Mareels (2023) at 99.96%.This outstanding performance is complemented by a lower average loss of 0.0002, proving its accuracy and flexibility in detecting malware. The ability of the GN-BiLSTM model to maintain high accuracy proves its usability and reliability in classifying a diverse variety of malware samples and its better capabilities when compared to previous techniques.

The GN-BiLSTM achieved an impressive accuracy of 85.48% in classifying malware categories. This performance surpasses other cutting-edge techniques, including 85.04% by Roy et al. (2023), 82.21% by Abualhaj et al. (2024), and 84.56% by Shafin, Karmakar & Mareels (2023). The ability to classify malware samples into specific categories is a result of the GN-BiLSTM model’s excellent feature extraction and learning capabilities. Furthermore, the GN-BiLSTM had the lowest average loss of 0.3553 among the examined models, confirming its efficiency in reducing classification errors and strengthening its power in category-specific malware classification.

In classifying malware families, the GN-BiLSTM performed exceptionally with an accuracy of 74.68% which is higher than the findings of Abualhaj et al. (2024) (66.93%), Roy et al. (2023) (70.29%), and Mezina & Burget (2022) (83%). The model’s ability to identify malware categories accurately proves its outstanding learning and generalization abilities. Furthermore, by comparing with other techniques, the GN-BiLSTM has the lowest average loss of 0.4896, proving its effectiveness in reducing misclassification rates and correctly detecting malware families.

The comparison of these related works shows the importance of the practical application of a GN-BiLSTM-based approach for the detection and classification of categories and families of obfuscated malware. Identifying the correct malware category and family improves the defense mechanism and helps to decrease analysis time. Results showed that the proposed model not only can detect, classify, and attribute malware families with the highest accuracy and low loss when compared to other works but is also implemented and checked on real-world latest malware families to verify its application. Results on the self-created dataset indicate that the proposed model can be implemented on real-world samples and will produce higher results in detecting, classifying, and attributing obfuscated malware.

The performance of deep learning models in different operating environments is an important factor to consider when deploying them in actual applications. To examine the adaptability of the proposed model, further experiments are conducted on Linux-based operating systems, and hardware configurations ranging from high-end servers to resource-constrained edge devices. The proposed model maintained acceptable detection and classification accuracy with reduced training time in all scenarios, processing performance and resource utilization varied depending on hardware and system characteristics. These findings highlight the importance of adapting the model to different situations to effectively classify ransomware in a range of real-world scenarios.

We also executed a series of high-traffic simulations to test the scalability of the proposed model for larger datasets and real-time detection. Samples from old and new datasets are utilized to assess the ability of the model to analyze large volumes of data. Real-time detection is also performed on some latest ransomware samples to demonstrate its ability to detect ransomware in near-real time while maintaining accuracy, loss, and speed. The results show that, while the GN-BiLSTM model scales well with larger datasets, only minor memory and processing time changes are required to retain performance under high-load conditions. These scalability enhancements make the model perfect for use in real-time ransomware detection systems that must process huge amounts of data quickly and correctly.

Limitations

Despite its exceptional performance, the proposed model has some limits. The datasets used in this study have a limited number of ransomware samples and may limit the model’s generalizability because of its breadth and diversity. In real-world scenarios, novel ransomware families and complicated obfuscation tactics may be inaccurately represented, reducing the model’s ability to detect previously unreported samples. Furthermore, real-time detection presents a challenge since complex models such as CNN-LSTM and CNN-BiLSTM may have longer processing times, limiting their applicability in systems that demand rapid responses. While the models performed well in the experimental setup, their performance in other operational situations, such as alternative operating systems, hardware, or network circumstances, has not been properly tested. Furthermore, the models’ scalability for bigger datasets or real-time systems is unknown, because ransomware detection systems must handle enormous data volumes effectively. Though the datasets contain obfuscated malware, new evasion tactics may outperform present detection capabilities, demanding frequent model upgrades.

Conclusion and future work

With the invention of obfuscation techniques to save programming codes, malicious actors misused them and adopted those techniques to make ransomware undetectable. Static, signature-based, and pattern-based approaches with machine learning have failed to detect, classify, and attribute advanced obfuscated ransomware. Deep learning-based methods have proven helpful in detection by analyzing obfuscated ransomware in depth. A novel, group-normalized BiLSTM (GN-BiLSTM) base model for ransomware detection, classification, and family attribution is proposed. The OMM-2022 dataset is used to train six DL models (CNN, MLP, LSTM, CNN-LSTM, CNN-BiLSTM), including the proposed method GN-BiLSTM. Our technique achieved 99.99%, 85.48%, and 74.65% accuracy in detecting, classifying, and attributing ransomware, respectively. To validate the proposed method in real-world implementation, a new dataset of 21,752 records (including 11 ransomware families) and 10,876 benign samples is created to train the models. Experimental results depicted that the proposed scheme achieved 99.20% accuracy in detection, 97.44% in classifying category, and 96.23% in attributing its family. Consequently, the proposed GN-BiLSTM technique can not only detect ransomware but also classify new variants with high accuracy when compared to previous work. In the future, we plan to implement the proposed method in detecting ransomware in other fields like the Internet-of-Things (IoT), Android, and iOS operating systems. We also plan to increase the malware categories and families in the dataset to further validate the effectiveness of the proposed approach.

Supplemental Information

Supplemental Information 1 Ransomware Dataset.

Additional Information and Declarations

Competing Interests

Author Contributions

Data Availability

Khursheed Aurangzeb is an Academic Editor for PeerJ.

Amjad Hussain conceived and designed the experiments, performed the experiments, performed the computation work, prepared figures and/or tables, and approved the final draft.

Ayesha Saadia conceived and designed the experiments, performed the experiments, performed the computation work, prepared figures and/or tables, authored or reviewed drafts of the article, and approved the final draft.

Musaed Alhussein analyzed the data, authored or reviewed drafts of the article, preparation of Paper for Submission in PeerJ Journal, and approved the final draft.

Ammara Gul performed the experiments, authored or reviewed drafts of the article, and approved the final draft.

Khursheed Aurangzeb analyzed the data, prepared figures and/or tables, authored or reviewed drafts of the article, preparation of Paper for Submission in PeerJ Journal, and approved the final draft.

The following information was supplied regarding data availability:

The CIC-MalMem-2022 dataset published by the Canadian Institute for Cybersecurity, University of New Brunswick is available at: https://www.unb.ca/cic/datasets/malmem-2022.html.

The self-created dataset and the detection, category-wise classification, and family attribution code, is available at GitHub and Zenodo:

- https://github.com/khbdevelopers/Enhancing-Ransomware-Defense-Detection-and-Classification-of-Ransomware.

- Amjad Hussain, A. H. (2024). Ransomware Dataset 2024 (1.0) [Data set]. Zenodo. https://doi.org/10.5281/zenodo.13890887.

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
