# Peer review of "Enhancing ransomware defense: deep learning-based detection and family-wise classification of evolving threats"

_PeerJ Computer Science, doi:10.7717/peerj-cs.2546_

## Round 0.1 · original submission · Major Revisions

Dear authors,

Thank you for submitting your article. Based on reviews' comments, your article has not yet been recommended for publication in its current form. However, we encourage you to address the concerns and criticisms of the reviewer and to resubmit your article once you have updated it accordingly. Reviewer 1 has advised you to provide specific reference. You are welcome to add it if you think it is useful and relevant. However, you are under no obligation to include it, and if you do not, it will not affect my decision. Furthermore, explanations of the equations should be checked. All variables should be written in italic as in the equations. Their definitions and boundaries should be defined. Please use equation numbers for referencing the equations. Do not use "as", "below", “following” “as follows”, etc. Necessary references should also be provided for relevant equations. Many of the equations are part of the related sentences. Attention is needed for correct sentence formation.

Best wishes,

Reviewer 1 ·

Basic reporting

Ransomware, which locks or encrypts files for ransom, has become harder to detect due to obfuscation techniques. Traditional machine learning methods have struggled with advanced variants, but deep learning has shown promise in both detection and classification. This work introduces a Group Normalization-based Bidirectional Long Short-Term Memory (GN-BiLSTM) model to improve accuracy in detecting, classifying, and attributing ransomware families. Validated against the CIC-MalMem-2022 and a self-collected dataset of 6000 samples, the GN-BiLSTM model achieved impressive accuracy rates of up to 99.99%. This approach outperforms others and shows potential for real-world application.

The paper is well-organized in terms of structure, but there is room for improvement in language and grammar. Please check the comments below.
- The language of the abstract can be improved. I suggest the following revisions:
Ransomware is a type of malware that locks access to or encrypts its victim’s files for a ransom to be paid to get back locked or encrypted data. With the invention of obfuscation techniques, it became difficult to detect its new variants. Identifying the exact malware category and family can help to prepare for possible attacks. Traditional machine learning-based approaches failed to detect and classify advanced obfuscated ransomware variants using existing pattern-matching and signature-based detection techniques. Deep learning-based approaches have proven helpful in both detection and classification by analyzing obfuscated ransomware deeply. Researchers have contributed mainly to detection but minimal (minimally) to family attribution. This research aims to address all these multi-class classification problems by leveraging the power of deep learning. … (For the rest, I suggest to either use the present or the past tense – and not both)
- Line 60: typo  compared to
- Paragraph starting line 55: "- 'Crypto' and 'locker' should either both be capitalized or both lowercase. Additionally, I suggest sticking to the past tense and avoiding the use of multiple tenses (for example, see line 73)."
- Paragraph starting line 78: same comment about the tense
- I suggest including a sentence or two early in the introduction that outlines the high-level target of the paper. Although the contribution is stated clearly later on, the reader has to wait too long to understand the paper's aim, at least at a high level.
- Speaking of the contributions:
The first one: The first contribution is to propose a DL-based method that can detect the latest obfuscated ransomware variants with higher accuracy: I suggest to use the word high or otherwise explain what exactly is meant by ‘higher’ accuracy.
The second one: The second contribution is to perform multi-class classification to detect malware categories and its family. (or their families?)
The third one: The third contribution is to validate the proposed approach by collecting the latest real-world ransomware samples, creating dataset and applying proposed approach to it. I suggest to improve the writing here.
- Line 138: typo  The fifth section provides the conclusion and possible future research directions from this research.
- I believe the authors provide sufficient background literature related to their work. Also, the literature review is clear and well-stated. However, more (recent) surveys could be added to provide additional context. For instance: “The Age of Ransomware: A Survey on the Evolution, Taxonomy, and Research Directions”.
- Line 118: reference must be after the period.
- Line 124: use the present tense and stick to the present in this paragraph.
- Line 127: Benigns
- Line 128: CVS format (abbreviation used without defining)
- Line 276: for our proposed..
- Line 278: feature selection
- Figure 1 is blurry. I suggested to fix it.
- Figure 4 is not readable.
- Lines 390 - 391, “In this research, correlation and mutual information feature importance techniques are implemented on both datasets to check feature relations.”
- "In general, a thorough revision is needed to correct all grammatical mistakes, including punctuation, tense, and capitalization errors."
- Line 416: remove comma after Ha and sentence is not clear.
- Line 417: are they both from 1 to 1?
- Line 455: Comma after “Traditionally”
- Figures 9, 10, 11, 12, 13, 14, 15 and 16 are blurry.

In acknowledgements: no need to capitalize Research.

Experimental design

The experimental design involved training and evaluating six deep learning models—CNN, MLP, LSTM, CNN-LSTM, CNN-BiLSTM, and the proposed GN-BiLSTM—using the CIC-MalMem-2022 and OMM-2022 datasets, which consist of obfuscated malware samples. Additionally, a self-collected dataset of 6000 ransomware samples from 10 families and 6000 benign samples was used to further validate the models. The models were trained to detect, classify, and attribute ransomware, with the GN-BiLSTM model showing superior performance across all metrics.
I believe the experimental design is comprehensive and well-structured, effectively demonstrating the capabilities of the GN-BiLSTM model. However, I suggest incorporating a broader range of ransomware samples to enhance the robustness of the findings. Additionally, a more detailed analysis of the model's performance in various operating environments and its scalability would provide valuable insights for practical deployment and ensure its applicability in real-world scenarios.

Validity of the findings

The validity of the findings is supported by the robust performance of the GN-BiLSTM model across multiple datasets, including the CIC-MalMem-2022, OMM-2022, and a self-collected dataset, achieving accuracy rates of up to 99.99% in detection. The consistency of high accuracy in both controlled and real-world scenarios underscores the model's reliability. Furthermore, the inclusion of multiple deep learning models for comparison validates the superior performance of the GN-BiLSTM model, reinforcing the study's conclusions about its effectiveness in detecting and classifying obfuscated ransomware.

Additional comments

The paper introduces a new model that significantly improves how we detect and categorize complex ransomware. It's impressive to see it tested across various models and datasets, including a set of ransomware samples. The results are robust, achieving up to 99.99% accuracy in detecting ransomware and 85.48% in classifying its type. This highlights its practical application in addressing real cybersecurity challenges.
However, the paper could benefit from clearer presentation and analysis of its results. Enhancing the clarity of these sections is crucial for meeting the journal's publication standards. Additionally, providing a more detailed explanation of the model's architecture would help readers better understand its novelty and effectiveness. Furthermore, expanding the discussion on the limitations of the proposed approach could enrich the paper's contribution.

Reviewer 2 ·

Basic reporting

1. All figures are blurred therefore it is required to redraw it.
2. Require to check typos and grammar.
2. Some tables and figures need to be rearranged so that they should not be in the reference section. Check the same for all the tables and figures.

Experimental design

1. The abstract needs to be reframed to align with the current industry problem caused by malware.
2. The authors outlined the merits and drawbacks of a few similar works in the literature review section. Rather than just listing the shortcomings of each study, it is recommended that the writers raises the common issues raised by this research and go into great depth about how the suggested solution addresses each one.
3. Kindly check line no 310 as I have found some error in description.
4. There is a need to show algorithm for necessary all modules in the proposed methodology section.
5.6. The authors have conducted several tests in the experiment section to confirm the effectiveness of the suggested system; nevertheless, the results lack credibility and support because certain important cause analysis was not done.
6. Kindly discuss the scenario in the case on new type of rensomare attack happen for existing approach.
7. For evident results I think sample size is too small for existing approach.

Validity of the findings

1. Require to mention limitation for existing approach.

---

## Round 0.2 · accepted · Accept

Dear Authors,

Thank you for the revision. The reviewers have accepted the paper, and I concur that it has been sufficiently revised and is suitable for publication. However, during the production stage, it would be prudent to give due consideration to the recommendation regarding sentence tense put forth by Reviewer 1.

Best wishes,

Reviewer 1 ·

Basic reporting

1. The comment below was given in the previous review as well.
In the abstract: “Researchers have contributed mainly to detection but minimal (minimally) to family attribution.”
The abstract tense is still not consistent.

All of the remaining comments have been addressed and the paper is now ready.

Experimental design

The modifications have been made as suggested.

Validity of the findings

The modifications have been made as suggested.

Reviewer 2 ·

Basic reporting

No comment

Experimental design

No comment

Validity of the findings

No comment

Additional comments

No comment